# Stereotactic Body Radiotherapy (SBRT) for the Treatment of Primary Localized Renal Cell Carcinoma: A Systematic Review and Meta-Analysis

**DOI:** 10.3390/cancers16193276

**Published:** 2024-09-26

**Authors:** Agata Suleja, Mateusz Bilski, Ekaterina Laukhtina, Tamás Fazekas, Akihiro Matsukawa, Ichiro Tsuboi, Stefano Mancon, Robert Schulz, Timo F. W. Soeterik, Mikołaj Przydacz, Łukasz Nyk, Paweł Rajwa, Wojciech Majewski, Riccardo Campi, Shahrokh F. Shariat, Marcin Miszczyk

**Affiliations:** 1III Department of Radiotherapy and Chemotherapy, Maria Skłodowska-Curie National Research Institute of Oncology (MSCNRIO), 44-102 Gliwice, Poland; 2Department of Radiotherapy, Medical University of Lublin, 20-059 Lublin, Poland; 3Brachytherapy Department, Saint John’s Cancer Center, 20-090 Lublin, Poland; 4Radiotherapy Department, Saint John’s Cancer Center, 20-090 Lublin, Poland; 5Department of Urology, Comprehensive Cancer Center, Medical University of Vienna, 1090 Vienna, Austria; 6Institute for Urology and Reproductive Health, Sechenov University, 119992 Moscow, Russia; 7Department of Urology, Semmelweis University, 1083 Budapest, Hungary; 8Department of Urology, Jikei University School of Medicine, Tokyo 105-8461, Japan; 9Department of Urology, Shimane University Faculty of Medicine, Izumo 693-8504, Japan; 10Department of Urology, Humanitas Clinical and Research Institute IRCCS, 20089 Milan, Italy; 11Department of Biochemical Science, Humanitas University, 20072 Milan, Italy; 12Department of Urology, University Medical Center Hamburg-Eppendorf, 20251 Hamburg, Germany; 13Department of Radiation Oncology, University Medical Center, 3584 Utrecht, The Netherlands; 14Department of Urology, Jagiellonian University Medical College, 30-688 Krakow, Poland; 15Second Department of Urology, Centre of Postgraduate Medical Education, 01-813 Warsaw, Poland; 16Department of Urology, Medical University of Silesia, 40-800 Zabrze, Poland; 17Radiotherapy Department, Maria Skłodowska-Curie National Research Institute of Oncology (MSCNRIO), 44-102 Gliwice, Poland; 18Department of Experimental and Clinical Medicine, University of Florence, 50139 Florence, Italy; 19Unit of Urological Robotic Surgery and Renal Transplantation, University of Florence, Careggi Hospital, 50134 Florence, Italy; 20Karl Landsteiner Institute of Urology and Andrology, 1010 Vienna, Austria; 21Department of Urology, Second Faculty of Medicine, Charles University, 15006 Prague, Czech Republic; 22Division of Urology, Department of Special Surgery, University of Jordan, Amman 11942, Jordan; 23Department of Urology, Weill Cornell Medical College, New York, NY 10065, USA; 24Department of Urology, University of Texas Southwestern, Dallas, TX 75390, USA; 25Research Centre for Evidence Medicine, Urology Department, Tabriz University of Medical Sciences, Tabriz 5166/15731, Iran; 26Collegium Medicum—Faculty of Medicine, WSB University, 41-300 Dąbrowa Górnicza, Poland

**Keywords:** kidney cancer, radiation, CyberKnife, non-metastatic, clear cell carcinoma, definitive treatment

## Abstract

**Simple Summary:**

Renal cancer is the eighth most frequent cancer in Europe, and its prevalence is increasing. Surgery is the treatment of choice for localised renal cell carcinoma requiring interventional management, but less invasive treatment methods are emerging. Stereotactic body radiotherapy (SBRT) utilises precise delivery of high doses of radiation to ablate the primary cancer. In our systematic review and meta-analysis, we pooled data from available prospective trials, including 13 studies involving 308 patients. The results of the meta-analysis show that SBRT for localised renal cell carcinoma is highly effective in controlling local diseases and has low complication rates. In the second year, 97% of patients were free from local recurrence. Only 3% experienced severe adverse events, which included abdominal pain and fatigue. SBRT presents a valuable treatment for patients who require treatment but cannot undergo surgery; however, it has not been yet confirmed to be equieffective to surgery as trials directly comparing these methods are missing.

**Abstract:**

Context: Surgery is the gold standard for the local treatment of primary renal cell carcinoma (RCC), but alternatives are emerging. We conducted a systematic review and meta-analysis to assess the results of prospective studies using definitive stereotactic body radiotherapy (SBRT) to treat primary localised RCC. Evidence acquisition: This review was prospectively registered in PROSPERO (CRD42023447274). We searched PubMed, Embase, Scopus, and Google Scholar for reports of prospective studies published since 2003, describing the outcomes of SBRT for localised RCC. Meta-analyses were performed for local control (LC), overall survival (OS), and rates of adverse events (AEs) using generalised linear mixed models (GLMMs). Outcomes were presented as rates with corresponding 95% confidence intervals (95% CIs). Risk-of-bias was assessed using the ROBINS-I tool. Evidence synthesis: Of the 2983 records, 13 prospective studies (*n* = 308) were included in the meta-analysis. The median diameter of the irradiated tumours ranged between 1.9 and 5.5 cm in individual studies. Grade ≥ 3 AEs were reported in 15 patients, and their estimated rate was 0.03 (95%CI: 0.01–0.11; *n* = 291). One- and two-year LC rates were 0.98 (95%CI: 0.95–0.99; *n* = 293) and 0.97 (95%CI: 0.93–0.99; *n* = 253), while one- and two-year OS rates were 0.95 (95%CI: 0.88–0.98; *n* = 294) and 0.86 (95%CI: 0.77–0.91; *n* = 224). There was no statistically significant heterogeneity, and the estimations were consistent after excluding studies at a high risk of bias in a sensitivity analysis. Major limitations include a relatively short follow-up, inhomogeneous reporting of renal function deterioration, and a lack of prospective comparative evidence. Conclusions: The short-term results suggest that SBRT is a valuable treatment method for selected inoperable patients (or those who refuse surgery) with localised RCC associated with low rates of high-grade AEs and excellent LC. However, until the long-term data from randomised controlled trials are available, surgical management remains a standard of care in operable patients.

## 1. Introduction

According to the GLOBOCAN 2022 data, renal cell carcinoma (RCC) is the eighth most common cancer diagnosed in Europe, and its incidence is increasing [1,2]. RCC is often found incidentally [3,4] and primarily affects older individuals. Resection is the gold standard of tissue diagnosis and treatment for clinically localised disease [5,6,7]. However, not all individuals are optimal surgical candidates due to the typically late onset of the disease, often associated with (age-related) comorbidities [8].

Active surveillance (AS) is an option for elderly patients, especially those with impaired kidney function and small renal masses. Many of these patients do not require surgical resection of the tumour [9,10,11,12], but those who experience disease progression on AS may eventually require an interventional treatment. Another approach involves minimally or non-invasive focal therapies that preserve renal function, including cryoablation (CA) and radiofrequency ablation (RFA), which should only be offered to well-selected patients with adequately located small tumours (distant from bowel, urinary tract, and major vessels) [13,14]. There is growing evidence that stereotactic body radiotherapy (SBRT) could also be considered in selected inoperable patients with more advanced diseases for local disease control [5].

RCC is historically considered a radioresistant malignant disease, but its resilience to radiation can be overcome by high doses per fraction used in SBRT [15]. Improved image guidance and respiratory motion compensation make it, indeed, possible to administer ablative doses to the tumour while limiting the damage to nearby organs at risk (OARs). Major advances in software and hardware solutions in treatment planning and delivery techniques have established SBRT as a safe and efficient method for patients unfit for surgery. However, the quality of the evidence was low, comprising mostly data from retrospective studies and prospective non-comparative trials [5,16]. More recently, after the success of the proof-of-concept studies, well-designed prospective trials that assess the clinical outcomes of SBRT in primary RCC patients have been performed. Our systematic review and meta-analysis aim to synthesise the emerging high-quality data and provide a comprehensive understanding of the evidence (and its quality) supporting the use of SBRT in the clinical management of patients with localised RCC.

## 2. Evidence Acquisition

### 2.1. Search Strategy

The prospective registration of the study in PROSPERO (CRD42023447274) was performed. The review was conducted according to the Preferred Reporting Items for Systematic Reviews and Meta-analyses (PRISMA), and the checklists for PRISMA 2020 are available in Appendix A [17]. The research question was developed following the Population, Intervention, Comparator, Outcome, and Study Design (PICOS) framework (Appendix A). We searched MEDLINE (via PubMed), Embase (via Ovid), and Scopus to identify records published between 2003/01/01 and 2023/07/21. The search was updated on 2024/04/09, prior to the initiation of the statistical analysis. The top 200 hits were retrieved from Google Scholar during the first search. A detailed description of the search strategy can be found in Appendix A. The screening process was performed using Rayyan software (https://www.rayyan.ai/, Qatar Computing Research Institute, Doha, Qatar) [18]. Two investigators conducted independent abstract screening, followed by the retrieval and assessment of full-text publications. Selected full-text articles were subject to backward citation searching. If multiple records reported results from the same trial, the most recent was selected. Disagreements were resolved through consensus mediated by co-investigators.

### 2.2. Study Selection

We included studies investigating clinical outcomes and safety in patients with localised RCC (Population) treated with definitive SBRT using fraction doses of at least 5 Gy (Intervention) in single-arm trials or compared with any standard-of-care systemic therapy (Comparator). We retrieved studies that reported local control (LC), renal function preservation, progression-free survival (PFS), overall survival (OS) and rates of grade ≥ 3 adverse events (AEs) (Outcomes) based on data from prospective trials only (Study design; PICOS framework). Studies investigating palliative treatments, metastases-directed therapies, combinations of SBRT with other local and/or systemic therapies, and salvage treatments of previously treated lesions were excluded from the analysis. Multifocal disease, previous RCC treatments of other lesions (such as contralateral nephrectomy), and minor subgroups of N1/M1 patients were not considered an exclusion criterion. Study results published only as conference abstracts and non-English language reports were excluded.

### 2.3. Data Extraction

Data on study details, patient and treatment characteristics, and outcome measures were independently extracted by two investigators. Conflicts were resolved through mediation by a third investigator. The outcome measures included one-, two-, and three-year OS, LC, and PFS, post-treatment decline in estimated Glomerular Filtration Rate (eGFR), and rates of treatment-related grade ≥ 3 AEs according to the Common Terminology Criteria for Adverse Events (CTCAE) v3.0–v5.0 [19]. The LC was assessed using the Response Evaluation Criteria in Solid Tumors (RECIST; version 1.1) [20], except for one study in which the assessment method was not specified [21]. WebPlotDigitizer v.4.6 software was used to retrieve data from the figures [22].

### 2.4. Risk of Bias Assessment

Each study was evaluated independently by two investigators using Cochrane Collaboration’s ROBINS-I tool for non-randomised studies [23]. Conflicts were resolved through mediation with co-investigators.

### 2.5. Statistical Analysis

For the purpose of the meta-analysis, the rates of treatment-related grade ≥ 3 AEs and probabilities of time-to-event outcomes (OS, LC, and PFS) were analysed as proportions. In the case of AE rates, the numerator was the number of patients experiencing a given endpoint. For OS, the LC numerator was obtained by multiplying the probability at a given time estimated with the Kaplan–Meier method and the total number of analysed patients rounded to the nearest integer. The denominator was the total number of patients evaluated for a given outcome. If not provided in the manuscript, extracted individual patient data were used to calculate the probabilities and/or rates.

The meta-analysis was performed using the generalised linear mixed model (GLMM) with the logit transformation as the measure of effect size, using the metaprop function in the metafor package with the method set to “GLMM” and the summary measure set to ”PLOGIT”. Confidence intervals for individual study results were calculated as exact binomial (Clopper–Pearson) intervals and with a Hartung–Knapp adjustment. The logit-transformed rates were back-transformed to probabilities to facilitate the interpretation of forest plots, which were used to present outcomes with 95% confidence intervals (95% CIs) for individual studies and meta-analytic averages.

Heterogeneity between studies was assessed using Cochran’s Q test (*p*-values < 0.05 considered significant). Publication bias was assessed using funnel plots. In cases where 10 or more studies were included in an analysis, Peters’ linear regression test was used to formally assess plot asymmetry. Statistical analyses were performed using R software v4.3.2 (R Foundation for Statistical Computing, Vienna, Austria), and RStudio, including “survival”, “meta”, and “metafor” packages [24]. *p*-values < 0.05 were considered significant. All tests were two-sided.

## 3. Evidence Synthesis

### 3.1. Study Selection and Characteristics

Among the 2225 screened individual study records, we identified 13 reports of individual prospective single-arm studies published between 2015 and 2024. These reports primarily described the results of SBRT for localised RCC, including 308 patients with a total of 312 renal lesions. The PRISMA flow diagram is presented in Figure 1. Five trials were conducted in Europe (*n* = 114; 37%) [21,25,26,27,28], four in the United States (*n* = 66; 21%) [29,30,31,32], two in Asia (*n* = 21; 7%) [33,34], one in Australia (*n* = 37; 12%) [35], and one study was multicentric (*n* = 70; 23%) [36]. Ten studies reported the final trial outcomes (*n* = 233; 75%) [21,25,27,29,30,31,33,34,35,36], while one presented initial results (*n* = 20; 7%) [32], and two reported data from prospective registries (*n* = 55; 18%) [26,28]. No controlled studies were identified.

The median age of the patients was above 70 years in all except two included trials (*n* = 47; 15%) [25,27], and the majority of the patients were male (*n* = 221; 72%). The median diameter of the treated tumours ranged between 1.9 and 5.5 cm in the included studies. The majority of the lesions were histopathological confirmed (*n* = 274; 88%). Out of 248 evaluated patients, the most prevalent clinical stages were T1a (*n* = 123; 50%) and T1b (*n* = 105; 42%). The majority of patients received C-arm-based radiotherapy (*n* = 166; 54%), followed by CyberKnife (*n* = 78; 25%), MR-guided radiotherapy (MRgRT) (*n* = 56; 18%), and carbon ion radiotherapy (CIRT) (*n* = 8; 3%). The basic details of study populations and interventions are summarised in Table 1.

### 3.2. Risk of Bias Assessment

The summary of the RoB analysis and its applicability concerns are presented in Appendix A. Eleven studies (*n* = 287; 93%) were assessed as having a moderate risk of bias, and two studies (*n* = 21; 7%) were assessed as having a serious risk of bias according to the ROBINS-I tool. In both cases, the concerns pertained bias in the classification of the intervention, with no clear definition of intervention groups.

### 3.3. Oncological Outcomes

Data on the 1-year LC were available for all included studies, for a total of 293 patients. As presented in Figure 2, the estimated 1-year LC was 0.98 (95% CI 0.95–0.99), and the estimates reported in individual studies ranged from 0.92 to 1. The 2-year LC was available for 253 patients included in 12 studies. The estimated 2-year LC rate was 0.97 (95% CI 0.93–0.99) and ranged from 0.89 to 1 in individual studies. The 3-year LC was available for 179 patients included in eight studies. The estimated 3-year LC rate was 0.95 (95% CI 0.81–0.99) and ranged from 0.75 to 1 in individual studies. There was no evidence of significant heterogeneity for any of the LC analyses.

Data on 1-year OS were available for all the included studies, for a total of 294 patients. As presented in Appendix A, the estimated 1-year OS was 0.95 (95% CI 0.88–0.98) and ranged from 0.79 to 1 in individual studies. The data on 2-year OS were available for 224 patients included in 12 studies. The estimated 2-year OS was 0.86 (95% CI 0.77–0.91) and ranged from 0.63 to 1 in individual studies. The 3-yr OS was available for 220 patients included in nine studies. The estimated 3-year OS rate was 0.78 (95%CI 0.67–0.86) and ranged from 0.61 to 0.89 in individual studies. There was no evidence of significant heterogeneity for any of the OS analyses.

### 3.4. Toxicity

The incidence of treatment-related G3 AEs reported for each trial accounted for a total of 291 patients. As presented in Figure 3, the pooled estimated rate was 0.03 (95% CI 0.01–0.11), and there was no evidence of significant heterogeneity. Seven studies (*n* = 139; 48%) reported no grade ≥ 3 Aes. In total, fourteen G3 and one G4 event were reported. Detailed descriptions of the Aes of interest are presented in Appendix A [21,25,26,27,28,29,30,31,32,33,34,35,36].

### 3.5. Renal Function

Due to the heterogeneity in the reporting of renal function preservation, a meta-analysis was omitted. The median baseline eGFR ranged from 44.5 mL/min to 83.7 mL/min. In five studies, the authors assessed the renal function decline at one year [27,28,31,35,36], which ranged from −8.7 mL/min to −11 mL/min. The data extracted on various estimators of renal function preservation are summarised in Appendix A [21,25,26,27,28,29,30,31,32,33,34,35,36].

### 3.6. Funnel Plots and Sensitivity Analyses

The funnel plots for applicable analyses are summarised in Appendix A. There was no evidence of statistically significant publication bias. Additionally, we performed a sensitivity analysis by excluding studies at high RoB and re-calculating the primary outcomes of the meta-analysis (LC, rate of G ≥ 3 AEs). The results are presented in Appendix A. The pooled rate of G ≥ 3 AEs remained low at 0.03 (95% CI 0.01–0.12), while the 1-, 2-, and 3-year LC rates remained high, at 0.98 (95% CI 0.94–0.99), 0.98 (95% CI 0.92–0.99), and 0.95 (95% CI 0.74–0.99), respectively.

## 4. Discussion

In this study, we summarised the outcomes of prospective single-arm trials evaluating the efficacy of SBRT for primary RCC, which represents the highest quality of the currently available evidence. There are several important findings in our study. First, we identified that the treatment is associated with high rates of local control over the course of available follow-up. Second, the rates of high-grade AEs are low, and we did not identify any substantial safety signals in the available studies. Finally, although the heterogeneity did not allow for a quantitative synthesis, the majority of the included studies reported relatively minimal decreases in the eGFR, confirming that SBRT can be considered a form of nephron-sparing therapy. The available evidence suggests that SBRT for localised RCC presents a valuable treatment option; however, data from randomised controlled trials comparing it with standard-of-care surgery are missing; therefore, it should be primarily considered as a treatment option for non-surgical candidates who are not candidates for AS.

Although the majority of the included studies utilised commonly available intensity-modulated RT delivery methods, using C-arm linear accelerators, several alternatives have been explored. In particular, robotic CyberKnife systems [25,28,29,30], adaptive MRgRT [26,32], and CIRT [34] were used in the included studies. However, as effective local treatments appear safe and feasible using any of the tested RT methods, including C-arm-based conformal or intensity-modulated RT [21,27,28,31,33,35,36], it does not appear that more sophisticated technologies are necessary. On the other hand, the generalisability to other RT delivery methods is important for the accessibility of treatment and could allow for more precise SBRT in technically difficult cases.

There is significant heterogeneity in RT schedules used in the included trials, with doses ranging from a single fraction of 25 Gy [25] to 72 Gy in 12 fractions [34]. A recent meta-analysis by Huang et al. found no significant association between the oncological outcomes and the RT dose [37]. It could be associated with the fact that within each of the available prospective trials, sufficient RT doses were delivered to achieve local response, but it also could be related to the generally latent biology of these tumours. The prior hypothesis is supported by the consistently high rates of 1- and 2-year LC and low toxicity rates confirm the feasibility of investigated RT regiments. However, there is insufficient evidence to confirm that lower RT doses, even within the range of investigated fractionation schemes, do not affect long-time LC. More follow-up is necessary to identify optimal dose fractionation schedules.

The majority of patients included in the trials had T1a or T1b stage disease; however, some studies included more advanced tumours (> 7 cm) and still achieved almost perfect LC at one year [26,30,34]. We believe that this points towards an opportunity to extend SBRT to RCC patients with more advanced disease stages. At the same time, toxicity remains an important concern, and it is unlikely that SBRT will be applicable as a sole method of treatment for particularly large RCCs. There is no comparative data except one pilot study published as a conference abstract, which compared SBRT to invasive thermal ablation. Due to the small number of participants (*n* = 21), and major deviations from the intended randomisation goal, no comparative conclusions can be drawn at this moment. Nevertheless, SBRT appeared feasible, although patients might require more time to achieve a complete pathological response compared to those treated with RFA [38].

There were several trials investigating the results of dose escalation [21,29,30,34]. The 2-year LC rates were comparable to other studies. This could be associated with a relatively short follow-up, and improvements in LC could manifest later. While the majority of patients qualified for SBRT due to RCC are elderly, dose escalation can be valuable for younger patients with an otherwise long-term expected survival. At the same time, it should be remembered that a minor improvement in LC could be offset by increased toxicity. Additionally, it is not clear whether a single high dose is better than fractionated SBRT, unless this is addressed in a prospective randomised study [37,39]. A particular case of an uncommon yet interesting dose escalation concept is a focal boost to positive lymph nodes. There was one patient presenting N+ disease in the study by Siva et al. [36]. This could be a potential future research direction, as radiotherapy is generally a well-recognised treatment method for regional lymph node metastases in abdominal and pelvic cancers.

There are limited data on quality of life (QoL) in patients undergoing SBRT in primary RCC. Zarkar et al. reported that they collected such data and that the results would be published in subsequent papers [28]. Out of 308 patients evaluated in this meta-analysis, 45 were reported to have a solitary kidney. These patients could have particularly improved long-time QoL by preserving renal function and delaying the need for dialysis. It is expected that the effect for the patient would be similar to that of a partial nephrectomy, with the exception that it could be used in patients contraindicated to receive surgical management [40]. Furthermore, patients undergoing SBRT are often elderly, with significant comorbidities and impaired kidney function. Therefore, their eGFR is likely to decline regardless of the oncological treatment [41], and the nephron-sparing effect can be underestimated in available studies.

While our results confirm the observations from previously pooled analyses [16,42,43,44], restricting the analysis to prospective studies reduces the vulnerability towards typical limitations of retrospective data collection. However, there are several limitations to our study. First, the limited duration of the follow-up does not allow for the assessment of long-term outcomes. Second, heterogeneity in reporting renal injury did not allow for a quantitative assessment of this endpoint. Third, some of the patients treated with SBRT for RCC could also be candidates for AS, limiting the generalisability of the method. Fourth, histopathological confirmation of the lesion was not available in all cases, meaning that a small subset of patients could have been treated for non-malignant tumours, contributing to high LC. Fifth, as pointed out by Bertolo et al., the selection criteria were often subjective, including patients’ refusal of surgery. This puts into question whether evaluated patients would remain inoperable if consulted by unaffiliated physicians [45]. Most importantly, to date, no high-quality randomised controlled trials are available, and therefore there is insufficient data to assess the value of SBRT in the setting of patients who could be candidates for both SBRT and surgical management.

## 5. Conclusions

The use of SBRT for definitive local treatment in patients with primary localised RCC is supported by evidence from prospective trials. The treatment is associated with high efficacy in controlling local diseases, low rates of complications, and is feasible using standard C-arm SBRT-capable linear accelerators. However, to date, there is no evidence from randomised controlled trials that would allow for a direct comparison with standard surgical management, and long-term outcome data are limited.

While SBRT is a safe and efficient alternative which can be offered to inoperable patients or those who refuse surgery, we believe that the evidence is still insufficient to routinely use SBRT as an alternative to surgery in potential candidates for partial nephrectomy.

## Figures and Tables

**Figure 1 cancers-16-03276-f001:**
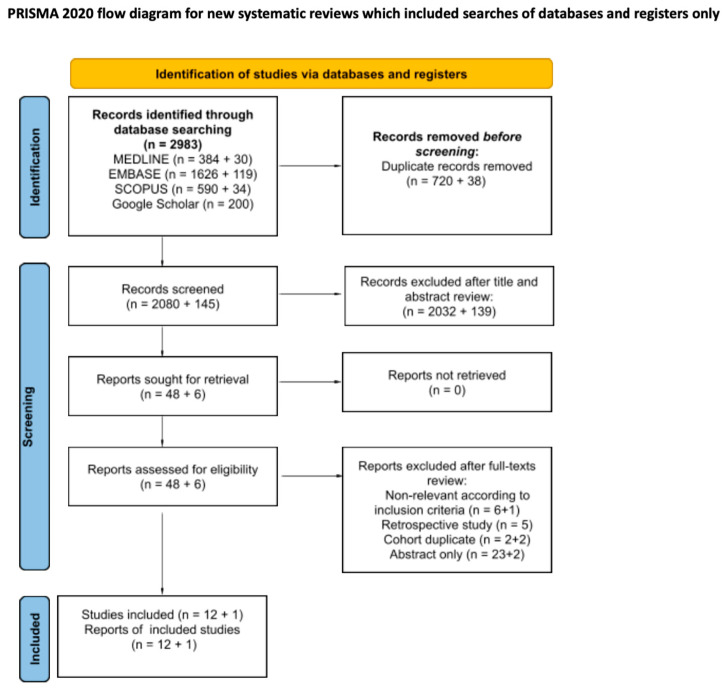
Preferred Reporting Items for Systematic Reviews and Meta-analyses (PRISMA) 2020 flowchart of study included in the systematic review.

**Figure 2 cancers-16-03276-f002:**
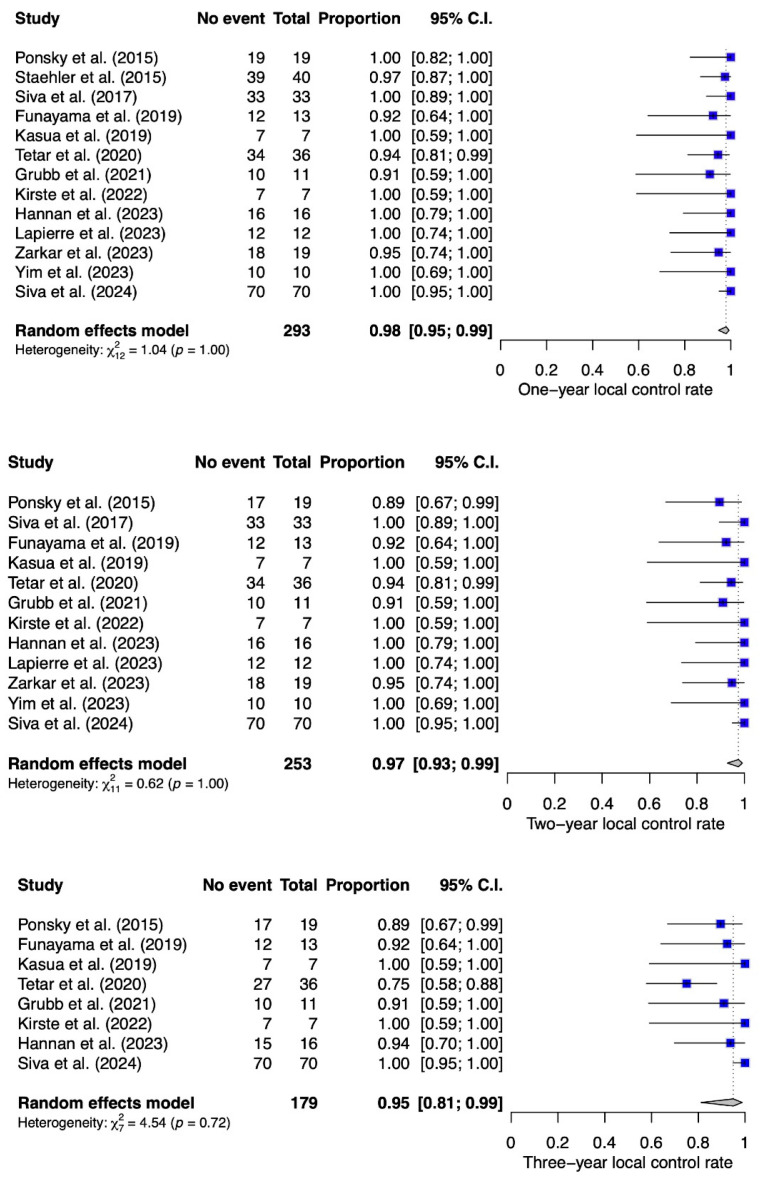
Local control following stereotactic body radiotherapy for clinically localised renal tumours at 1, 2 and 3 years following treatment [21,25,26,27,28,29,30,31,32,33,34,35,36].

**Figure 3 cancers-16-03276-f003:**
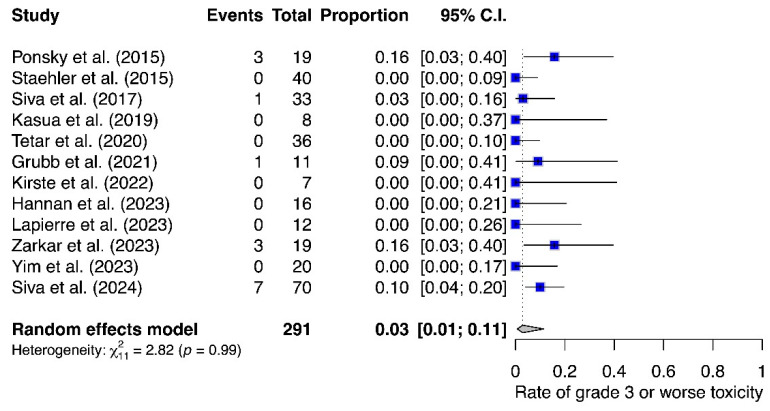
Rates of G3 or worse treatment-associated toxicity in patients treated with stereotactic body radiotherapy for clinically localised renal tumours [21,25,26,27,28,29,30,31,32,34,35,36].

**Table 1 cancers-16-03276-t001:** Basic characteristics of 13 prospective trials reporting data on outcomes of SBRT in patients with localised renal cell carcinoma.

Publication Year, 1st Author	Study Type	Radiotherapy Delivery Method	Fractionation Scheme (Dose [Gy]/Fractions (Numb. of Patients)	Number of Patients	Age (Median)	T Stage, *n* (%)	Lesion Diameter (Median)	HP Confirmed	Lesions Treated	Solitary Kidney	Median FU (mo)	Toxicity ≥3 (*n*)	Evaluated Clinical Outcomes
Ponsky et al. (2015) [29]	SA	CyberKnife	24/4 (4); 32/4 (6); 40/4 (3); 48/4 (6)	19	77.6	N/A	N/A	18	19	N/A	13.7	3	Overall survival, local control
Staehler et al. (2015) [25]	SA	CyberKnife	25/1 (40)	40	65.6	T1a: 37 (93%)	N/A	45	45	29	28.1	0	Local control, renal function, progression-free survival, overall survival
Siva et al. (2017) [35]	SA	3D-CRT	42/3 (17); 26/1 (17)	37	78	T1a: 13 (35%) T1b: 23 (62%) T2a: 1 (3%)	4.8	34	34	N/A	24	1	Local control, freedom from distant progression, overall survival
Funayama et al. (2019) [33]	SA	C-arm photon	60/10 (6); 70/10 (7)	13	72	T1a: 12 (92%) T1b: 1 (8%)	1.9	0	13	6	48.3 (mean)	N/A	Local control, overall survival
Kasuya et al. (2019) [34]	SA	CIRT	66/8 (5) or 72/12 (3) [RBE]	8	71	T1a: 3 (38%) T1b: 4 (50%) T3a: 1 (12%)	4.3	2	8	1	43.1	0	Local control, overall survival
Tetar et al. (2020) [26]	SA	Adaptive MRgRT	40/5 (36)	36	78.1	T1a: 5 (14%) T1b: 23 (64%) T2a: 8 (22%)	5.5	20	36	N/A	16.4	0	Local control, overall survival, freedom form any progression
Grubb et al. (2021) [30]	SA	CyberKnife	48/3 (4); 54/3 (4); 60/3 (3)	11	76.5 (mean)	T1a: 7 (64%) T1b: 3 (27%) T2a: 1 (9%)	3.7	11	11	2	34.3	1	Overall survival, disease free survival, local control, freedom from distant metastases
Kirste et al. (2022) [27]	SA	VMAT	50/5 (6); 60/8 (1)	7	44	N/A	2.8	7	8	0	43	0	Local control, cancer-specific survival, progression-free-survival, overall survival
Hannan et al. (2023) [31]	SA	VMAT	36/3 (10); 40/5 (6)	16	72	T1a: 13 (81%) T1b: 3 (19%)	3.2	16	16	1	36	0	Local control, overall survival, progression-free survival
Lapierre et al. (2023) [21]	SA	3D-CRT	32/4 (3); 40/5 (3); 40/4 (3); 48/4 (3)	12	78	N/A	3.3	12	12	0	23	0	Local control
Zarkar et al. (2023) [28]	SA	CyberKnife VMAT	26/1 (8); 42/3 (11)	19	76	N/A	4.5	19	20	1	17	3	Local control, overall survival
Yim et al. (2023) [32]	SA	Adaptive MRgRT	40/5 (20)	20	79.5	T1a: 9 (45%) T1b: 9 (45%) T2a: 1 (5%) T3a: 1 (5%)	4.4	20	20	5	17	0	Local control
Siva et al. (2024) [36]	SA	3D-CRT (8), IMRT (6), VMAT (56)	N/A	70	77	T1a: 24 (34%) T1b: 39 (56%) T2a: 6 (9%) T3a: 1 (1%)	4.6	70	70	N/A	43	7	Local control, overall survival, cancer-specific survival, freedom from distant metastases

Abbreviations: 3D-CRT—three-dimensional conformal technique; CIRT—carbon ion radiotherapy; FU—follow-up; HP—histopathological; MRgRT—MR-guided radiotherapy; N/A—not applicable or not available; RBE—relative biological effectiveness; SA—single-arm trial; SBRT—stereotactic body radiotherapy, VMAT—volumetric modulated arc therapy.

## Data Availability

Data used in the manuscript are publicly available.

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
