# Peer review of "Stereotactic Body Radiotherapy (SBRT) for the Treatment of Primary Localized Renal Cell Carcinoma: A Systematic Review and Meta-Analysis"

_cancers, 2024, doi:10.3390/cancers16193276_

Round 1

Reviewer 1 Report

Comments and Suggestions for Authors

It is an excellent systematic review and meta-analysis evaluating the outcomes of SBRT for localized renal cell carcinoma. I would like to congratulate the authors for the excellent study design. The methodology and the interpretation of the results are adequate. All the main points are described in the discussion section. 

Comment 1: The flow diagram and the risk of bias should be transferred to the main manuscript.

Comments on the Quality of English Language

As mentioned above

Author Response

Comment 1: The flow diagram and the risk of bias should be transferred to the main manuscript.

Response 1: 

Dear Reviewer,

Thank you for your valuable comment. We have added the PRISMA 2020 flow diagram to the main manuscript as suggested. However, due to space limitations, we were unable to include the risk of bias figure. We appreciate your insightful feedback and your efforts to improve our manuscript.

Sincerely,
Authors

Reviewer 2 Report

Comments and Suggestions for Authors The reported data are very interesting and methodologically well analyzed and presented. However, I consider some revisions to be appropriate:

Line 85: the reasons why resection is the gold standard of tissue diagnosis of renal cancer as well as for treatment should be briefly detailed

Line 95: "adequately located" tumors should be better explained (i.e. far from vascular bundles?)   Line 101: I suppose do the authors mean ablative doses to the tumor mass?   Line 102: "Scientific evolution of SBRT" minimizes the strong, revolutionary impact of SBRT on cancer treatment. Advances in treatment planning and delivery techniques established SBRT as a proper alternative for patients considered unfit for surgery.   Line 209-2010: The description of the applied/detected method for LC assessment (RECIST; 209 version 1.1 except for one study) should be moved to the section "Data extraction".   Beyond this, I think the discussion lacks considerations on radiation doses and fractionations. The relative radioresistance of kidney tumors is described in the Introduction, moreover the authors cited trials investigating RT dose escalation (Line 287), but an extreme heterogeneity in RT schedules is reported in Table 1 (from 60 Gy/3, or single-dose which is definitely ablative due to biologically effective dose (BED) >>100, to 24 Gy/4 or 32/4, 60/10, or radiosurgery 26 Gy/1). I suggest integrating the Discussion with this topic. Likewise, a few more details on SBRT technique (inhomogeneous dose distribution, focusing high doses per fraction on the target volume and contemporary sparing of the surrounding healthy tissues, ...) should be desirable. The authors stated (Line 272) "it does not appear that more sophisticated technologies are necessary", but planning non-homogeneous dose distributions is necessary in optimizing the efficacy/toxicity ratio of SBRT, even performed with conformal techniques. Comments on the Quality of English Language

A moderate revision of the English version of the manuscript is desirable. For instance:

Line 92: an interventional treatment ; Line 99: high doses per fraction ; Line 179: "Among the 2,225 screened individual study records" is more appropriate; Line 181: there is an extensive use of the word "comprising", I suggest replacing it with "including" sometimes ; Line 222 all the included studies ; Line 231: consider revising the english version of the whole sentence "The incidences of treatment-related grade ≥3 AEs were retrieved for each trial", i.e. the incidence of treatment-related G3 AEs reported for each trial accounted for a total of 291 patients" ; Line 247: there was no evidence of...publication bias ; Line 255: which represents the highest quality of the currently available evidence ; Line 267: although the majority ; ....

Author Response

Comments and Suggestions for Authors

The reported data are very interesting and methodologically well analyzed and presented. However, I consider some revisions to be appropriate:

Response: Dear reviewer,

Thank you for your insightful review and your contribution to improve our manuscript. You can find answers to your comments below. 

Comment 1: Line 85: the reasons why resection is the gold standard of tissue diagnosis of renal cancer as well as for treatment should be briefly detailed

Response 1: We thank the Reviewer for this comment. We have cited NCCN, EAU, and ESMO guidelines, which all state that surgery is a first-choice treatment.

Comment 2: Line 95: "adequately located" tumors should be better explained (i.e. far from vascular bundles?)  

Response 2: We thank the Reviewer for this comment. We have revised the manuscript to provide a clearer explanation of “adequately located” tumors. The sentence now reads: “adequately located (distant from the bowel, urinary tract, and major vessels) small tumors.”

Comment 3: Line 101: I suppose do the authors mean ablative doses to the tumor mass?  

Response 3:  We thank the Reviewer for this comment. Yes, we are referring to ablative doses targeting the tumour mass. We have revised the manuscript to explicitly state that we mean the tumour mass in this context to avoid any confusion.

Comment 4: Line 102: "Scientific evolution of SBRT" minimizes the strong, revolutionary impact of SBRT on cancer treatment. Advances in treatment planning and delivery techniques established SBRT as a proper alternative for patients considered unfit for surgery.  

Response 4: We thank the Reviewer for this valuable insight. We agree that we may have underestimated the impact of SBRT in our original phrasing. We have revised the sentence in the manuscript, which now reads: “Major advances in treatment planning and delivery techniques, supported by innovative software and hardware solutions, have established SBRT as a safe and effective alternative for patients unfit for surgery.”

Comment 5: Line 209-2010: The description of the applied/detected method for LC assessment (RECIST; 209 version 1.1 except for one study) should be moved to the section "Data extraction".  

Response 5: We thank the Reviewer for this valuable comment. We have moved the description of the method for LC assessment (RECIST version 1.1, except for one study) to the “Data extraction” section, as suggested.

Comment 6: Beyond this, I think the discussion lacks considerations on radiation doses and fractionations. The relative radioresistance of kidney tumors is described in the Introduction, moreover the authors cited trials investigating RT dose escalation (Line 287), but an extreme heterogeneity in RT schedules is reported in Table 1 (from 60 Gy/3, or single-dose which is definitely ablative due to biologically effective dose (BED) >>100, to 24 Gy/4 or 32/4, 60/10, or radiosurgery 26 Gy/1). I suggest integrating the Discussion with this topic.

Response 6: We thank the Reviewer for this insightful comment. We agree that the radiation doses and fractionation schedules in the studies we reviewed exhibit significant variability, as outlined in Table 1. However, a recent meta-analysis on the dose-response relationship for localized renal cell carcinoma by Huang et al. found no significant association between oncological outcomes and the radiation dose applied. The regimens used in the included studies align with those recommended by Huang et al. (25-26 Gy/1 or 42-48 Gy/3), in each of the available prospective trial, sufficient RT dose was delivered, achieving local response. However, the local control rates assessed in these studies are typically reported at 1- and 2-year, which is relatively short There is insufficient evidence to confirm that lower doses over a longer period would be equally effective. We have integrated this important consideration into the discussion, acknowledging the heterogeneity in SBRT schedules. We have integrated this important point into the discussion to acknowledge the heterogeneity in SBRT schedules: “ There is significant heterogeneity in RT schedules used in the included trials, with doses ranging from a single fraction of 25 Gy [25] to 72 Gy in 12 fractions [34]. A recent meta-analysis by Huang et al. found no significant association between the oncological outcomes and the RT dose [37]. It could be associated with the fact that within each of the available prospective trials, sufficient RT doses were delivered to achieve local response, but also could be related to the generally latent biology of these tumours. The prior hypothesis is supported by the consistently high rates of 1- and 2-year LC, and low toxicity rates confirm the feasibility of investigated RT regiments. However, there is insufficient evidence to confirm that lower RT doses, even within the range of investigated fractionation schemes, do not affect long-time LC. More follow-up is necessary to identify optimal dose fractionation schedules.”

Comment 7: Likewise, a few more details on SBRT technique (inhomogeneous dose distribution, focusing high doses per fraction on the target volume and contemporary sparing of the surrounding healthy tissues, …) should be desirable. The authors stated (Line 272) “it does not appear that more sophisticated technologies are necessary”, but planning non-homogeneous dose distributions is necessary in optimizing the efficacy/toxicity ratio of SBRT, even performed with conformal techniques.

Response 7: We thank the Reviewer for this comment; however, we would like to point out that at this point, there is no evidence to support superiority of any of the tested RT regimens, some of which included relatively non-sophisticated solutions. We believe that the feasibility of performing RCC SBRT on most modern SBRT-capable systems is a strong aspect of the method, and would prefer to refrain from making statements emphasizing necessity of any particular aspects. We do believe that the SBRT needs to be prescribed and reported according to the ICRU91 standards, which necessitate certain level of precision, and permit high degree of inhomogeneity (which we personally use, especially with CyberKnife); however, we do not feel that there is sufficient evidence to indicate that e.g. inhomogeneity in dose distribution should be maximised, at least not beyond what is specified in the ICRU anyways.

Comments on the Quality of English Language

Comment 8: A moderate revision of the English version of the manuscript is desirable. For instance:

Line 92: an interventional treatment ; Line 99: high doses per fraction ; Line 179: "Among the 2,225 screened individual study records" is more appropriate; Line 181: there is an extensive use of the word "comprising", I suggest replacing it with "including" sometimes ; Line 222 all the included studies ; Line 231: consider revising the english version of the whole sentence "The incidences of treatment-related grade ≥3 AEs were retrieved for each trial", i.e. the incidence of treatment-related G3 AEs reported for each trial accounted for a total of 291 patients" ; Line 247: there was no evidence of...publication bias ; Line 255: which represents the highest quality of the currently available evidence ; Line 267: although the majority ; ....

Response 8: We thank the reviewer for valuable comments regarding the quality of the English language. We have incorporated all of the suggested improvements into the manuscript, including the specific revisions noted, and these can be found in the revised version.

Reviewer 3 Report

Comments and Suggestions for Authors

The interest of this review is that it includes patients with small tumors (T1a_T1B) treated exclusively with radiotherapy, since there are meta-analyses of combined treatments with systemic therapy.

The data on toxicity are not very extensive and are not analyzed in depth, which is attributable to the lack of information in the referenced articles.

I believe that the abstract could be improved by mentioning that the review only includes tumors treated exclusively with radiotherapy.

Author Response

Comment 1: 

I believe that the abstract could be improved by mentioning that the review only includes tumors treated exclusively with radiotherapy.

Response 1: 

Dear Reviewer,

Thank you for your valuable feedback. We have revised the abstract to clearly indicate that SBRT was the only treatment method used in the studies included in our review.

Sincerely,
Authors